# Relationships between PTSD severity and suicidal ideation in Cambodian women: The role of individual symptom clusters, trauma types and depression

## Research Article

suicide; post-traumatic stress; trauma; low-income countries

**Corresponding author:**
Nil Ean;
Email: nilean@yahoo.com

Kathryn Fleddermann[1] , Steven E. Bruce[1], Robert H. Paul[2] and Nil Ean[3,4]

[1]University of Missouri-St. Louis, St. Louis, MO, USA; [2]Missouri Institute of Mental Health, St. Louis, MO, USA; [3]Royal University of Phnom Penh, Phnom Penh, Cambodia and [4]The Center for Trauma Care and Research Organization, Phnom Penh, Cambodia

## Abstract

Suicidal ideation and trauma exposure are significant health challenges worldwide, and their interaction increases their burden on individuals and communities. However, limited research has been devoted to these conditions in low- and middle-income countries, where the majority of the burden of these disorders exists. Additionally, unique cultural factors that may contribute to differential relationships in these symptoms and disorders make this an important area to explore. This study examines relationships between the number and types of adverse exposures, PTSD symptoms and severity, depression and suicidal ideation in a sample of Cambodian women with experiences of trauma using logistic and linear regressions. Overall, PTSD severity significantly contributes to suicidal ideation, with hyperarousal symptoms playing a particularly influential role in this association. Further, adverse experiences, including physical abuse and parental mental health problems, contributed significantly to increased suicidal ideation. Lastly, depression severity partially mediates the relationship between PTSD severity and suicidal ideation. These results illustrate the significant role of PTSD in the experience of suicidal ideation, particularly within regions like Cambodia with high trauma loads. These findings point to psychological constructs that may be especially important to include in suicidality screening tools and to target within prevention and intervention efforts.

## Impact statement

It is well-established that PTSD can increase the likelihood of experiencing suicidal ideation, but research on this relationship in areas that bear the heaviest burden of these disorders is limited. Low- and middle-income countries experience an outsized proportion of traumatic stress and suicidality in comparison to high-income countries, but the vast majority of research on these disorders has been completed in high-income areas. Different cultural views and expectations, as well as more frequent experiences of some types of trauma, may impact the relationships between these disorders and may point to a need to employ different screening and intervention techniques in these areas. This study sought to examine how PTSD severity and symptoms, types of adverse experiences and depression contribute to the development of suicidal ideation in Cambodia, a low-income country that carries significant trauma burdens. Results indicated that PTSD severity contributes to increased suicidal ideation, and hyperarousal symptoms seem to play a particular role in this. Individuals meeting diagnostic criteria for PTSD are at particularly heightened risk, though the relationship was significant even at lower levels of PTSD severity. Additionally, experiences of physical abuse and parental mental health problems contributed to increased suicidal ideation, but other trauma types like sexual abuse and neglect that are commonly associated with suicidal ideation in high-income areas did not contribute. These results point to constructs that may be especially useful to include in suicidality screening tools, as individuals with clinically significant PTSD, heightened hyperarousal symptoms and/or certain trauma types are at heightened risk of suicidal ideation in these areas. This also informs potential treatments that focus on PTSD symptomatology as a means to reduce suicidal ideation, particularly in areas where cultural factors like stigma may make directly treating suicidal ideation less acceptable.



## Introduction

Suicide presents a significant health burden globally, and considerable effort has been devoted to understanding risk factors that contribute to increased suicidal ideation and action. The contributions of traumatic events and resulting PTSD to suicidal ideation have been frequently studied,

but little research has focused on these factors and their connections in lower-resourced areas (Knipe et al., 2022; Lovero et al., 2023). While many of these relationships may be similar across cultures and global regions, numerous cultures exhibit culture-bound syndromes and symptoms of PTSD, as well as different types of events that may be traumatic, that are unique and not seen in Western regions (Hinton et al., 2012; Agger, 2015). Because of this, examining the relationships between PTSD and suicidal ideation in understudied regions is vital to creating a more thorough understanding of the mechanisms and development of suicidal ideation in these populations and overall.

### Suicidal ideation

Suicide accounts for a large portion of preventable deaths that occur each year, and despite over 80% of deaths by suicide occurring in low- and middle-income countries (LMICs), only 15% of research on suicide is done in these regions (Knipe et al., 2022; Lovero et al., 2023). It is estimated that suicide rates in the Southeast Asian region exceed the global rate, and this difference is particularly pronounced among women, with rates exceeding those in other regions (Naghavi, 2019; World Health Organization, 2021; Ilic and Ilic, 2022). Stemming from a lack of research attention to these global regions, estimates of suicidal ideation (wishing to die or that you were dead) also vary widely, with most research estimating lifetime rates between 15 and 25% with heightened rates in women (Nock et al., 2012; Jordans et al., 2018; Lovero et al., 2023).

Across global regions, there are significant risk factors for increased suicidal ideation, including comorbid psychiatric disorders, such as depression and anxiety, trauma exposures and resulting post-traumatic stress disorder (PTSD), and interpersonal factors, like low socioeconomic status, engaging in sex work and low educational attainment (Devries et al., 2011; Sahle et al., 2022; Lovero et al., 2023). These factors can individually increase suicidal ideation while also interacting with each other to create additional risk. Understanding these relationships is made more complicated by the significant comorbidity in disorders that individuals experience, along with the complex interactions of environmental and personal factors that can occur (Marshall et al., 2001; Galatzer-Levy et al., 2013). Particularly in low-resource environments, individuals can be exposed to unique and significant stressors that can appreciably raise their risk of suicidal ideation, beyond what may be seen in Western samples. This is the case in Southeast Asia and Cambodia in particular, with rates of past 12-month suicidal ideation in women estimated at 5.5%, which may be driven by factors like high levels of poverty, illness and lack of access to psychiatric care in this region (Sakisaka et al., 2018; Seponski et al., 2019a; Seponski et al., 2019b). However, more research is needed to fully understand how known risk factors for suicidal ideation contribute to its development in this region, due to the complexity of conditions there.

### Post-traumatic stress disorder (PTSD)

One significant contributor to the development of suicidal ideation is PTSD. Roughly 4% of the global population will meet diagnostic criteria for PTSD in their lifetime, although reported rates also vary significantly by region (Benjet et al., 2016; Kessler et al., 2017; Koenen et al., 2017; Levey et al., 2018). However, exposure to traumatic events is much higher, with recent global surveys estimating lifetime exposure rates to at least one traumatic event at roughly 70% (Benjet et al., 2016; Kessler et al., 2017). In LMICs,

these rates are even higher, with estimates ranging from 85 to 90%, though stigma around these experiences may contribute to underreporting (Seedat et al., 2004; Scott et al., 2013; Benjet et al., 2016; Levey et al., 2018). Global research suggests that those with any trauma exposure experience an average of two different trauma types and 4.6 traumatic incidents, though this too varies widely by country (Mollica et al., 1999; Benjet et al., 2016; Burchert et al., 2017; Kessler et al., 2017; Armes et al., 2019).

Importantly, not every trauma exposure will lead to the development of PTSD, but experiences of interpersonal traumas, especially sexual violence, are often considered the most likely to do so (Bruce et al., 2001; Wanklyn et al., 2016; Kessler et al., 2017). Conditional risks of developing PTSD after a trauma exposure vary significantly depending on trauma type, with odds ratios typically between 2.0 and 2.5 (Bruce et al., 2001; Agorastos et al., 2014). Additionally, trauma effects are likely additive, so individuals with multiple trauma exposures may be more likely to eventually develop PTSD (Agorastos et al., 2014). Sociodemographic factors like being female, having a lower income and having lower levels of education are also associated with a greater risk of lifetime PTSD development (Atwoli et al., 2015; Koenen et al., 2017; Seponski et al., 2019b). In LMICs, lifetime estimates of PTSD range from 25 to 35%, likely stemming from higher rates of exposure to traumatic events in general, and particularly because exposure to traumas associated with higher odds of developing PTSD are more common in these areas (de Jong et al., 2003; Steel et al., 2009).

The relationship between PTSD and suicidal ideation is relatively well-established, but the mechanisms and reasons for this are varied. Global research has found significant associations linking PTSD to increased risk of suicidal ideation, finding that a PTSD diagnosis leads to 2.7× greater odds of suicidal ideation in developing countries (Nock et al., 2009). Individual PTSD symptom clusters are also strongly related to the development of suicidal ideation, with the negative alterations in cognition and mood (NACM) and alterations in arousal and reactivity (AAR) clusters demonstrating the strongest relationship (Panagioti et al., 2015, 2017; Whiteman et al., 2021). Some evidence also implicates the intrusion cluster in the development of suicidal ideation, but this is less well-supported and has been found primarily in male populations, so more research is needed to understand these relationships in all-female and/or international populations (Davis et al., 2014; Smith et al., 2016; Watkins et al., 2017; Chu et al., 2021). Additionally, research suggests that most trauma types are associated with greater suicidal ideation, and that greater numbers of experienced traumas increase levels of ideation, with this being more pronounced in women than men (LeBouthillier et al., 2015; Fox et al., 2021). Taken together, this indicates a significant relationship between trauma exposure, PTSD and suicidal ideation, including in global contexts.

### Depression

Depression is the most common mental health disorder worldwide and is highly comorbid with many other psychiatric disorders, including PTSD (Lim et al., 2018; National Center for Health Statistics, 2018). Comorbidity of these disorders often causes more severe symptoms, greater impairment and worse outcomes as compared to individuals with only one disorder (Campbell et al., 2007; Ikin et al., 2010; Post et al., 2011; Wanklyn et al., 2016; Nichter et al., 2019; Klein Breteler et al., 2021). These relationships are also seen in international populations, with studies of refugees from Kenya and Bhutan finding that comorbidity was associated with

worse mental and physical health outcomes (Van Ommeren et al., 2002; Im et al., 2022).

Depression is also highly related to suicidal thoughts and actions. Prior studies suggest that odds ratios of future suicidal ideation in individuals with depression range from 5.1 to as high as 10.2, and these odds are higher for women than men (Nock et al., 2010; Veisani et al., 2017). Additional research suggests that this relationship is predicted not just by the existence of depression, but also by its severity, with odds of suicidal ideation increasing with increased depression severity (Gensichen et al., 2010; Chin et al., 2011; Farabaugh et al., 2012; Keilp et al., 2012; Wagenaar et al., 2012).

Because of the high rate of comorbidity between PTSD and depression, it is likely that there is also a combined contribution of these disorders to suicidal ideation, but the primary driver of this ideation is unclear. Some research indicates that depression alone is responsible for this increase, with depression significantly increasing suicidal ideation in individuals with PTSD (Bryan and Corso, 2011; Panagioti et al., 2012). Other research suggests that the comorbidity is the key, with significantly greater odds of suicidal ideation found among those with both disorders as compared to either alone (Cougle et al., 2009; Ramsawh et al., 2014; Veisani et al., 2017; Nichter et al., 2019). Little research on this has been completed with more diverse populations, emphasizing a need for greater understanding of the contributions of these disorders in other global regions.

## The effects of gender

Though PTSD is often examined primarily in male samples, due to increased trauma exposure in military samples and higher numbers of men in the military, there is evidence that women may have a higher risk for developing PTSD, along with higher rates of depression and a greater likelihood of developing comorbid depression and PTSD (Kessler et al., 1993; Weissman et al., 1993; Brewin et al., 2000; Salk et al., 2017; Shevlin et al., 2018; Im et al., 2022). Additionally, prior literature suggests women tend to experience different trauma types and may have more severe outcomes after trauma, so understanding if these factors contribute to differences in suicidal ideation is important (Clum et al., 2000; LeBouthillier et al., 2015; Wanklyn et al., 2016). Taken together, these factors point to a need to not just include more women in these areas of research, but to study women alone to determine if there are differences in this group.

## Cambodia

Cambodia is a particularly salient country to examine these relationships, as the traumatic history of the Khmer Rouge within Cambodia contributed to extremely high rates of trauma in the country. During the Khmer Rouge, an estimated two million people, or roughly 20–25% of the country's population, were killed and many more were subjected to torture, forced migration and forced labor (Reed and Keely, 2001; Hinton, 2009). These experiences contributed to extremely high rates of trauma, with up to 62% of refugees from the country reporting trauma, though true rates may be even higher given stigma around disclosing trauma and cultural differences in trauma definitions (Marshall et al., 2005). Additionally, extremely high rates of poverty, much of which is a result of the Khmer Rouge and its aftermath, continue to challenge much of the population in Cambodia (Van Schaack et al., 2011; Asian Development Bank, 2022). There is evidence that poverty can

increase the likelihood and severity of PTSD, depression and suicidal ideation, making LMICs like Cambodia even more important to study so that improved strategies for prevention and intervention can be developed for populations where these factors are prevalent (Lorant et al., 2003; Pan et al., 2013; Knipe et al., 2015; Herrera-Escobar et al., 2019; Raschke et al., 2022; Nillni et al., 2023). Other structural factors in Cambodia that may influence the presentation and impact of PTSD and other mental health disorders include lack of access to healthcare and ongoing community violence, making this an important area to examine the complex ways in which mental health challenges can manifest.

Cultural and social factors also make Cambodia particularly important to study, as concerns around issues like karma and stigma can significantly limit individuals' comfort or desire to report experiences of trauma or suicidal ideation (Han et al., 2013). The vast majority of Cambodians are Buddhist, which centers on ideas of karma and reincarnation, encouraging the belief that virtuous actions lead to positive outcomes and poor behavior causes negative outcomes. Strong beliefs in reincarnation, provided one has lived a virtuous life and followed the tenets of Buddhism, are also common (Buddhism, 2023). These values strongly influence Cambodian individuals' willingness to disclose experiences of trauma and/or suicidal ideation, as many may feel that having experienced trauma was deserved and was a karmic event, and they may be unwilling to discuss these events, as it may imply that they have done something bad in the past (Salinas and Salinas, 2021). Similarly, concerns with reincarnation and not wanting to shame one's family may prevent individuals from being willing to disclose suicidal ideation (Han et al., 2013; Hinton et al., 2013). Because of these factors, despite known high trauma loads within Cambodia, individuals are often unwilling to report these experiences or seek treatment for it due to concerns over how it will reflect on their family and themselves, what it means for their religious future and what others will think of them (Bertolote et al., 2005; Hinton et al., 2013).

Cultural factors may also impact the ways that these disorders manifest in Cambodian individuals, as many of the traditional symptoms and diagnostic criteria utilized in Western populations may be less salient in these groups. Beyond this, many symptoms and syndromes of PTSD seen in Cambodian populations are not seen in other populations and, therefore, may be missed or misunderstood if considering trauma from a purely Western perspective (Hinton et al., 2012; Agger, 2015). Cambodian individuals with PTSD often describe suffering from somatic complaints, including migraine-like symptoms, dizziness or light-headedness, chest pain or worry about heart attacks and sleep paralysis (Hinton et al., 2005, 2018, 2019). Though these may co-occur in Western conceptualizations of PTSD, they are central symptoms of PTSD in Cambodian individuals, demonstrating a key difference from research focused on Western populations, which requires further investigation to understand whether these symptoms impact the relationships seen with suicidal ideation.

Further, these cultural beliefs can extend to conceptualizations of suicide, as it is believed that individuals who died in unnatural or violent ways may be unable to pass on or may return as ghosts and lure others into danger. These "ghost attacks" are frightening for individuals who experience them, as the ghosts can cause harm to the individuals experiencing the attacks, leading to insanity, illness and even suicide for those individuals (Hinton et al., 2020; Hinton, 2021). Additionally, concerns about reincarnation cause significant stigmatization of suicidal ideation, as individuals who die by suicide will not be able to be reincarnated (Han et al., 2013). The complex

interactions of these religious and cultural beliefs around trauma and suicidal ideation may help to reduce the likelihood of individuals developing these disorders, as they do not fit within their belief systems, but the stigma created by this may also decrease support for individuals struggling and increase distress. This demonstrates the importance of studying the relationships between trauma and suicidal ideation, specifically within Cambodian populations.

### Current study

To develop a greater understanding of the relationship between PTSD and suicidal ideation in a Southeast Asian sample, this study examined the association between PTSD severity, individual symptom clusters and suicidal ideation within a female Cambodian population. We studied the strength of associations between PTSD severity and symptom cluster scores, suicidal ideation and trauma loading and type within this population to see whether relationships found in previous research with Western populations persist cross-culturally, along with examining the role of depression in PTSD and suicidal ideation severity.

## Methods

### Participants

A sample of 286 female Cambodian participants was included in this study. Secondary data from two datasets were combined to create a final sample. Dataset 1 includes 160 female entertainment workers recruited from four different provinces in Cambodia (Battambang, Siem Reap, Phnom Penh and Sihanoukville). Participants in this dataset met the following inclusion criteria: (1) Cambodian and able to speak Khmer language appropriately, (2) biological females who were at least 24 years old and (3) were able to give consent voluntarily.

Only female participants from Dataset 2 were eligible for inclusion in the final sample, leading to 183 out of 228 individuals being included from this sample. Missing data analysis was conducted on the main study measures using Little's test, and missing data were determined to be missing completely at random (Little, 1988). Fifty-seven participants from this sample with missing data on the main measures were removed from the final sample. This led to a final analysis sample of 126 from this sample.

Participants in Dataset 2 were recruited from psychological treatment organizations. Participants were self-referred for treatment or referred from nongovernmental organizations within or surrounding Phnom Penh or the Battambang province of Cambodia. Participants in this dataset met the following inclusion criteria: (1) elevated symptoms of PTSD and (2) no evidence of psychosis, organic brain disorder, cognitive impairment, dementia, acute suicidality or acute need for treatment. There were no age or gender requirements for this dataset.

For both datasets, a convenience sampling method was used to maximize sample size. Because of the challenges associated with accessing and studying these populations, particularly in relation to mental health problems, this method of sampling was most feasible. The data used for this secondary analysis were collected as part of a previous research study, which received ethical approval from the Vietnam National University (VNU) and the National Ethical Committee for Health Research in Cambodia (Approval Number: 054NECHR; Protocol: "Mental Health and Functioning in School Age Children of Female Entertainment Workers in Cambodia"). Informed consent was obtained from all research participants.

### Measures

#### Patient health Questionnaire-9 (PHQ-9)

The Patient Health Questionnaire-9 is a measure of depression that has been validated for use across treatment settings (Kroenke et al., 2001). The Khmer translation shows excellent internal reliability ($\alpha$ = .86) (Ean et al., 2019). It is a nine-question survey assessing symptoms from the past 2 weeks, with response options from 0 to 3. Question 9 of the PHQ-9 assesses current suicidal ideation, with 0 indicating no suicidal ideation up to 3 indicating suicidal ideation nearly every day. This question has been shown to be a robust predictor of suicidal ideation, with sensitivity between .84 and .90 and specificity between .69 and .74 (Uebelacker et al., 2011; Rossom et al., 2017; Kim et al., 2022)

A composite depression score for each participant was calculated. Because question 9 of the PHQ-9 was used to assess current suicidal ideation, only the first eight questions were summed to assess depression. Because of limited participants endorsing higher levels of suicidal ideation (2 or 3), question 9 of the PHQ-9 (assessing suicidal ideation) was collapsed into a dichotomous variable, with answers of 0 recoded as no suicidal ideation, while responses of 1–3 were recoded as experiencing suicidal ideation.

#### PTSD checklist for DSM-5 (PCL-5)

The PCL-5 is a commonly used measure to assess PTSD, validated for use in military, civilian and international populations (Weathers et al., 2013). It is a 20-item measure that assesses both overall symptom severity and severity of each of four symptom clusters. The PCL-5 has a clinical cutoff for overall symptom severity that can be used to establish a probable diagnosis of PSTD ($\geq$33 out of 80) (Weathers et al., 2013). The Khmer translation of the PCL-5 shows excellent internal reliability ($\alpha$ = .93) (Field and Chhim, 2008).

A total composite score for each participant was calculated by adding all 20 items together. Additionally, symptom cluster scores were calculated for each of the four PTSD symptom clusters (intrusion, avoidance, NACM and AAR) by summing the scores for the respective cluster items in the scale. All of these variables were treated as continuous for analysis. A dichotomous variable indicating whether or not a participant met diagnostic criteria for PTSD was created, with 0 indicating being below the cutoff and 1 indicating a probable diagnostic level of PTSD.

#### Adverse childhood experiences scale (ACEs)

To assess trauma loading and types, the Adverse Childhood Experiences scale (Felitti et al., 1998) was used, which quantifies the number and types of adverse experiences a person has experienced. Individuals indicate whether they experienced each of 10 possible events (emotional abuse, physical abuse, sexual abuse, emotional neglect, physical neglect, parental divorce/separation, witnessed domestic violence, parental substance use, parental mental illness and parental incarceration) before the age of 18 years. A total score can be calculated on a scale from 0 to 10 (with lower scores indicating fewer adverse experiences; Felitti et al., 1998). Individual adverse exposures can also be analyzed to assess relationships between particular adverse experiences and other outcomes. Prior analyses on this sample demonstrated adequate reliability of the Khmer version in this sample ($\alpha$ = .78) (Ean et al., 2019).

A total composite trauma score for each participant was calculated by adding the number of items endorsed together (total: 0–10). This composite score was treated as continuous for analysis. Responses to individual items are dichotomous variables (yes/no).

## Procedures

Data for dataset 1 were collected in person via a questionnaire. Participants were identified and contacted to participate through a local NGO. Three research assistants from the NGO branch office of each data collection site collected the data from participants, after receiving training on data collection, research ethics and procedures for supporting participants in distress. There was also an emotional support team, including two Master's degree-level psychologists, at each site. Interviews took place at each NGO site in a private room. Participants were compensated with a small stipend (equivalent to 5 USD) to cover transportation and time spent on the interview. The collected data were stored securely at the Royal University of Phnom Penh.

Data for dataset 2 were collected in-person via a self-report questionnaire during treatment sessions. All participants provided informed consent before participation.

Data cleaning and analysis were completed in SPSS Version 29.0.1.0 and R Version 4.4.1.

## Analysis

Descriptive analyses were completed for all measures. A series of logistic regressions was conducted to assess the relationships between overall PTSD severity, PTSD symptom cluster severities, number of reported traumas, trauma type and suicidal ideation. Multiple linear regression was used to assess the relationship between trauma type and PTSD severity. Mediated logistic regression was conducted to assess whether depression mediates the relationship between PTSD severity and suicidal ideation.

## Results

### Descriptive statistics and assumption checking

Independent samples $t$-tests were conducted to assess for differences between the two datasets on the main measures. The datasets were not significantly different from each other on any of these measures. Because of this, analysis was conducted with the datasets combined.

Descriptive statistics for primary measures are included in Table 1. Mean participant age was 32.05 years (SD = 7.75; range: 18–63). Overall, 53.8% of the total sample was above the diagnostic cutoff for PTSD. Among individuals who endorsed any suicidal ideation, 23.8% ($n = 68$) reported suicidal ideation "several days" in the past 2 weeks, 15.4% ($n = 44$) reported suicidal ideation "more

**Table 1.** Descriptive statistics for primary study measures

| | Mean | SD | Range of scores in the sample |
|---|---|---|---|
| PTSD severity | 33.94 | 3.13 | 0–79 |
| Intrusion severity | 8.40 | 5.07 | 0–20 |
| Avoidance severity | 3.51 | 2.45 | 0–8 |
| Negative alterations in cognition and mood severity | 11.85 | 7.27 | 0–27 |
| Alterations in arousal and reactivity severity | 10.18 | 5.82 | 0–24 |
| Total # of ACEs endorsed | 4.38 | 3.13 | 0–10 |
| Depression severity | 12.05 | 5.77 | 1–24 |

than half the days" in the past 2 weeks and 6.3% ($n = 18$) reported suicidal ideation "nearly every day" for the past 2 weeks. Because of the limited endorsement of more severe levels of suicidal ideation, a dichotomous variable was created, with individuals who endorsed any severity of suicidal ideation combined into one level. This resulted in 45.5% of the total sample endorsed experiencing suicidal ideation.

Assumptions for all regression models were met. No evidence of multicollinearity, nonlinearity, heteroscedasticity or non-normality of errors was found, so no corrections were necessary.

### PTSD and suicidal ideation

Results of a logistic regression indicate that endorsement of suicidal ideation depended significantly on PTSD severity. As PTSD severity increased, participants were significantly more likely to endorse suicidal ideation ($b = .065$, $p < .001$). Odds ratios suggest that for every 1-point increase in PTSD severity, participants were 1.07× more likely to endorse suicidal ideation (95% CI: 1.05–1.09). Furthermore, a logistic regression examining whether meeting PTSD diagnostic criteria (PCL-5 ≥ 33) affects the likelihood of suicidal ideation found that meeting diagnostic criteria was significantly associated with suicidal ideation ($b = 2.19$, $p < .001$), leading to 8.90× greater odds of suicidal ideation.

A multiple logistic regression assessing the relationship between PTSD symptom clusters and suicidal ideation found a significant relationship between the AAR symptom cluster and suicidal ideation ($b = .12$, $p = .002$), with a 1-point increase in AAR symptom severity associated with 1.13× greater odds of suicidal ideation (95% CI: 1.05–1.22). No other symptom clusters showed a significant relationship with suicidal ideation. Table 2 includes full regression results.

### Relationship between trauma types, PTSD and suicidal ideation

Overall, the number of ACEs an individual reported was significantly related to odds of endorsing suicidal ideation ($b = 0.14$, $p < .001$). Odds ratios indicate that for each additional trauma, participants were 1.15× more likely to endorse suicidal ideation (95% CI: 1.07–1.24). A multiple logistic regression assessing the relationship between individual ACEs and suicidal ideation indicated that physical abuse ($b = 1.08$, $p = .002$) and emotional neglect ($b = -0.86$, $p = .012$) were significantly related to increased endorsement of suicidal ideation. Full results for all adverse experiences are included in Table 3.

Further, a multiple linear regression model predicting overall PTSD severity from ACEs was also significant, $F(10, 275) = 4.90$, $p < .001$. Psychological abuse ($t(275) = 2.10$, $p = .037$) and parental mental health ($t(275) = 2.33$, $p = .021$) were significant predictors in the model, suggesting these particular trauma types increase PTSD symptomatology.

### Influence of depression and PTSD on suicidal ideation

A bootstrapped, mediated logistic regression was conducted, with PTSD severity as the predictor, suicidal ideation (dichotomous: yes/no) as the outcome and depression severity as the mediator. Results indicated that when controlling for depression, PTSD severity had a significant direct effect on the likelihood of endorsing suicidal ideation ($c' = 0.011$, SE = 0.012, $p < .001$, 95% CI: [0.006, 0.018]). The indirect effect of PTSD severity on suicidal ideation

**Table 2.** Results of a multiple logistic regression of the association between endorsement of suicidal ideation and PTSD symptom cluster severity

|                                              | *B*  | SE *B* | *p*  | OR   | 95% CI OR     |
|----------------------------------------------|------|--------|------|------|---------------|
| Avoidance                                    | 0.10 | 0.08   | .191 | 1.11 | [0.95–1.30]   |
| Intrusion                                    | 0.09 | 0.05   | .058 | 1.09 | [1.00–1.21]   |
| Negative alterations in cognition and mood   | 0.00 | 0.04   | .996 | 1.00 | [0.93–1.07]   |
| Alterations in arousal and reactivity        | 0.12 | 0.04   | .002 | 1.14 | [1.05–1.22]   |

**Table 3.** Multiple logistic regression table predicting odds of suicidal ideation from ACEs

|                                                    | *B*   | SE *B* | *p*   | OR   | 95% CI OR     |
|----------------------------------------------------|-------|--------|-------|------|---------------|
| Psychological abuse                                | −0.29 | 0.32   | 0.372 | 0.75 | [0.39–1.40]   |
| Physical abuse                                     | 1.08  | 0.35   | 0.002 | 2.95 | [1.50–5.91]   |
| Sexual abuse                                       | 0.56  | 0.32   | 0.080 | 1.75 | [0.93–3.27]   |
| Emotional neglect                                  | −0.88 | 0.34   | 0.010 | 0.42 | [0.21–0.80]   |
| Physical neglect                                   | 0.50  | 0.33   | 0.129 | 1.64 | [0.87–3.15]   |
| Parental separation or divorce                     | 0.17  | 0.28   | 0.541 | 1.19 | [0.68–2.04]   |
| Witnessed domestic violence                        | −0.20 | 0.34   | 0.545 | 0.82 | [0.42–1.57]   |
| Parental or other adult's substance abuse          | −0.20 | 0.32   | 0.523 | 0.82 | [0.43–1.52]   |
| Parental or other Adult's mental health problems   | 0.50  | 0.32   | 0.115 | 1.65 | [0.88–3.10]   |
| Parental or other adult's incarceration            | 0.12  | 0.31   | 0.706 | 1.12 | [0.61–2.07]   |

through depression severity was also significant ($ab$ = 0.005, SE = 0.009, 95% CI: [0.001, 0.008]). These findings suggest a partial mediation, in which PTSD severity is associated with the likelihood of endorsing suicidal ideation both directly and indirectly through increased depression severity.

## Discussion

Increasing understanding of the drivers of suicidal ideation is vital to improving prevention and early intervention strategies, particularly in relation to the complex interactions between sociocultural factors and mental health vulnerabilities. In a country like Cambodia, where the interplay between historical exposures to trauma and challenging social and economic conditions continues to significantly impact people's lives, understanding the relationship between trauma exposures and ongoing mental health concerns is key, though challenging. Particularly when considering the transition between generations who experienced the Khmer Rouge firsthand and those who have grown up in a region still working to recover from it, the need to better understand how early life adversity impacts adult mental health outcomes is vital to improving services for this population going forward. Additionally, women in Cambodia may be exposed to heightened stressors due to gender roles, expectations for them to care for their families and increased risks of gender-based violence. While this sample may have been more likely to be exposed to these risks than typical Cambodian women, due to their work in the entertainment industry and their increased mental health burden, many women in Cambodia contend with structural inequities that may contribute to similar risks and outcomes.

Prior studies in Cambodia have found lower rates of probable PTSD than were found in this sample, though studies with treatment-seeking populations and those with direct exposure to the Khmer Rouge have found much higher rates, so the combination of treatment-seeking and community-based individuals in this sample is likely responsible for this (Seponski, et al., 2019b; Maddock et al., 2023; Heuveline and Clague, 2024). However, because over half of the participants were above the diagnostic cutoff for PTSD, and this showed a significant relationship to suicidal ideation, these results suggest screening for suicidal ideation may be especially important for individuals with clinical levels of PTSD. However, a strong relationship was seen across all levels of PTSD severity, indicating that individuals who are exposed to traumatic events or are presenting for trauma screenings and treatments should also be screened for and provided with information regarding suicidal ideation, regardless of PTSD severity.

Additionally, the relationship of specific PTSD symptoms to suicidal ideation in this sample points to particular treatment targets that may be useful, as well as individuals to be targeted in prevention efforts. The significant relationship between the AAR symptom cluster and suicidal ideation in this sample aligns with some previous findings (Panagioti et al., 2017; Stanley et al., 2019; Morabito et al., 2020). However, the NACM cluster has also been predictive of suicidal ideation in prior research, though this was not true in our sample (Panagioti et al., 2015; Brown et al., 2018; Whiteman et al., 2021). These results suggest that the heightened reactivity and impulsivity associated with this symptom cluster are particularly distressing and more likely to lead to negative outcomes than other symptoms and that negative mood and thoughts are not as strongly involved in the development of suicidal ideation in this population.

This finding also helps to illuminate the connection between PTSD symptom clusters and suicidal ideation in different populations than have typically been researched. These results suggest that PTSD symptoms may differentially impact women in comparison

to men in this region, with AAR symptoms perhaps being more distressing in female populations. However, this association has been seen in other populations, so this relationship extends beyond only women (Panagioti et al., 2017; Stanley et al., 2019; Morabito et al., 2020). Notably, cultural presentations of distress in Cambodia can differ from Western conceptualizations, with more frequent experiences of somatic symptoms and culturally relevant descriptions of symptoms, such as "thinking too much" (Hinton et al., 2015). It is possible that hyperarousal symptoms more closely mirror these common cultural presentations and explanations of trauma symptoms, including insomnia, racing heartbeat and shortness of breath, which may help to explain why these symptoms are a source of particular distress in this population (Hinton et al., 2012, 2018). Taken together, these results suggest that individuals in Cambodia, and especially women in this population, presenting with heightened hyperarousal symptoms may be at higher risk for suicidal ideation and should be considered especially appropriate for screening and prevention resources. Additionally, it is possible that if addressing suicidal ideation directly within this population is not culturally appropriate due to stigma, treatment focused on hyperarousal symptoms could be a target for indirectly improving suicidal ideation.

The relationship of particular adverse experiences to PTSD severity and suicidal ideation is of particular interest, as exposures to sexual violence and other interpersonal traumas are typically thought to lead to the most severe negative outcomes, though this was not found in our sample (Wanklyn et al., 2016; Kessler et al., 2017). However, other interpersonal adverse experiences, including psychological and physical abuse, were associated with these outcomes, supporting the overall negative effect of interpersonal traumas. The significant relationship of parental mental health problems with PTSD severity is also unique, suggesting the important role that family dysfunction may play in increasing negative mental health outcomes for this population. Notably, given high rates of trauma exposure in older generations in Cambodia due to the Khmer Rouge, this particular adverse experience may be a proxy for intergenerational transmission of trauma that occurs within families (Field et al., 2013; Mak et al., 2021). Prior research has found evidence for transmission of trauma among families with parents who lived through the Khmer Rouge, via impacts on their parenting style and strategies for behavior management. However, there is also evidence for the transmission of increased resilience, as parents may provide their children with tools and strategies to manage distress that other populations may not have developed (Mak et al., 2021). Though it is not possible to know what parental mental health problems are being reported by this adverse experience item, more research is needed to better understand whether particular types of parental mental health problems are most strongly associated with later generations' own mental health difficulties and the process through which this occurs, as well as ways to foster transmission of resilience.

Overall, when considering these relationships, it is important to consider that there may be differences in the meaning of these adverse experiences shaped by cultural context. It is possible that experiences assessed in this measure have different meanings across cultures, and the impact of an adversity in one culture may be more significant than in another. While any of these experiences has the potential to be traumatic for an individual, some may be more common or viewed as more normal within a particular culture, so even if they are distressing, their effect may be less severe. Additionally, patriarchal structures in Cambodia may impact women's willingness to disclose these experiences, as cultural messaging may suggest that women will be "tarnished" if they have experienced a trauma, and that the experiences of trauma should be kept private (Brickell, 2008; Hanley et al., 2013). These belief systems may impact women's willingness to disclose particular types of events, but may also cause some adverse experiences to be particularly distressing if there is especially negative messaging about these experiences. Taken together, these factors may help explain why adverse events that are typically associated with negative adult outcomes were not in this population, though there will naturally be significant variation in these experiences across individuals, even within the same culture. However, it is also possible that certain adverse events are just differentially associated with these adult outcomes than have been seen in Western research contexts, and further research will be required to understand the underlying mechanism for these differences.

Better understanding specific adverse experiences that significantly contribute to severe outcomes like PTSD and suicidal ideation may allow for more targeted intervention and prevention efforts in this population. Individuals who are identified as having experienced these types of adverse experiences should receive particular attention for interventions for PTSD and suicidal ideation, as they are at higher risk for more severe outcomes. Additionally, these types of experiences may be areas to focus on in parenting and family interventions, as preventing them from occurring in the first place is vital. Lastly, the significant role that parental mental health appears to play in increasing PTSD severity also suggests that increasing overall access to mental health services is needed, though this is challenging to put into practice due to the limited availability of mental health care providers and services in Cambodia.

Lastly, the individual and combined contributions of PTSD and depression severity to suicidal ideation are of note, as the relative roles of PTSD and depression in the development of suicidal ideation remain under investigation (Cougle et al., 2009; Bryan and Corso, 2011; Panagioti et al., 2012; Ramsawh et al., 2014). The partial mediation found in this analysis suggests that the contribution of PTSD severity to suicidal ideation is more significant in this population than the added contributions of depression severity and supports a focus on PTSD prevention and interventions to reduce suicidal ideation for this group.

### Limitations

Due to limited research in international and all-female populations, not all measures that were used in this study have been validated for use in Cambodia yet, but most have been validated for use in international settings, including other low-income countries (Cholera et al., 2014; Ibrahim et al., 2018; Ho et al., 2019; Kidman et al., 2019; Phi et al., 2023). Additionally, all measures were translated and back-translated to best convey the meaning of survey items. Despite this, it is always possible that the meaning of different items does not translate well, linguistically or culturally, and that the interpretation of items may have been different for individuals in our sample. This study also did not address the possibility of cultural differences in symptom presentations that may exist in other global regions. While all measures showed strong reliability for the clinical constructs they assessed, it is possible that there are particular symptoms typical of PTSD in Cambodia that may have differentially driven these associations. Future research should examine these relationships using measures that assess cultural presentations of PTSD in order to assess the role of culture-bound syndromes in predicting suicidal ideation. Other limitations include retrospective evaluation of adverse experiences, which

may introduce error due to misremembering and may be affected by cultural stigma around experiencing traumatic events and desires to avoid bringing shame to oneself or one's family. Additionally, the use of a single question to assess for suicidal ideation, though this is commonly done, may not fully capture experiences of suicidal ideation. Further, as suicidal ideation is a stigmatized experience in Cambodia, reporting of suicidal ideation may have been reduced. Future projects expanding on this work may address these issues by adding more questions regarding suicidal ideation and including culturally appropriate ways of discussing suicidal ideation to increase comfort with endorsing this experience.

### Future research

Future research may seek to expand upon these findings by including increased demographic information, expanded measures of trauma exposures and measures of other mental health disorders, which may help to further conceptualize contributors to suicidal ideation in this population. Much remains unknown about what drives suicidal ideation in this group, but rates are particularly high compared to other populations, so delineating contributions of other disorders and environmental factors may improve understanding of the most effective prevention and intervention strategies. Additionally, incorporating qualitative data to further contextualize the perceptions and experiences of trauma and suicidality in this population will help to improve understandings of how these disorders present in Cambodia and how individuals understand and respond to these experiences.

### Conclusion

Suicidal ideation is a pressing public health burden worldwide, and many of the most under-resourced areas of the world bear the most significant weight of this. Despite this, little research has been devoted to understanding drivers of suicidal ideation in these areas of the world, due to logistical, financial and cultural barriers. In countries like Cambodia, where complex social and historical landscapes contribute to high prevalence of traumatic experiences and mental healthcare options are minimal, a better understanding of what drives these high rates of suicidal ideation is vital to increasing the availability of prevention and intervention options for these populations. Overall, PTSD severity drives suicidal ideation in this population, but certain adverse experiences interact with this relationship in complex ways, suggesting experiences that may be targets to screen for in individuals seeking care. These findings indicate that targeting individuals with high levels of trauma exposure and/or PTSD symptoms for early intervention and education regarding suicidal ideation may be particularly useful for reducing these symptoms going forward, but further research will be required to identify the most effective ways to address these areas moving forward.

**Open peer review.** To view the open peer review materials for this article, please visit http://doi.org/10.1017/gmh.2026.10195.

**Data availability statement.** The data that support the findings of this study are available from the authors upon reasonable request.

**Acknowledgments.** The research team would like to acknowledge the contributions of the research assistants and study therapists who supported this data collection, as well as the participants who shared their time and experiences.

**Author contribution.** Conceptualization: K.F., N.E., R.P., S.E.B.; Data Curation: N.E., Formal Analysis: K.F., Methodology: K.F., S.E.B.; Supervision: N.E., R.P., S.E.B.; Writing – Original Draft: K.F., Writing – Review and Editing: K.F., N.E., R.P., S.E.B.

**Financial support.** This research received no specific grant from any funding agency in the public, commercial or not-for-profit sectors.

**Competing interest.** The authors declare none.

**Ethics statements.** Ethical approval for this study was received from the Vietnam National University (VNU) and the National Ethical Committee for Health Research in Cambodia (Approval Number: 054NECHR; Protocol: "Mental Health and Functioning in School Age Children of Female Entertainment Workers in Cambodia"). Informed consent was obtained from all research participants.

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
