## [Reviewer Report]

Title: Relationships between PTSD severity and suicidal ideation in Cambodian women: The role of individual symptom clusters, trauma types, and depression

General Comments

This study addresses an important research gap by focusing on PTSD severity and its association with suicidal ideation among Cambodian women, a population often underrepresented in global mental health research. The findings are valuable, particularly as they shed light on the experiences of women from low-income countries in the South Asian region. Results suggest that PTSD severity significantly contributes to suicidal ideation, with hyperarousal symptoms showing a particularly strong association.

Overall, the Results and Discussion sections are clearly written and well-organized, effectively linking the findings to the research questions and relevant literature. However, there are several areas where the manuscript could be strengthened to improve clarity, methodological rigor, and readability.

Abstract

• The results sentence is incomplete:

“Overall, PTSD severity significantly contributes to suicidal ideation, with hyperarousal symptoms playing a particular individual role in this.”

Consider revising for clarity, e.g., “…with hyperarousal symptoms playing a particularly influential role in this association.”

In-Text Citation and Referencing

There are multiple inconsistencies in in-text citations and reference formatting. Please ensure all references are cited consistently according to the journal’s referencing style. Examples of issues include:

• Page 3-(DE Hinton et al. 2012); (Ilic and Ilic 2022); (World Health Organization 2021)

• Page 5-(MT Davis et al. 2014); (HL Smith et al. 2016)

• Page 7-(Reicherter and Eng 2011); (GN Marshall et al. 2005); (A Hinton 2009); (Reed and Keely 2001)

• Page 9- (Field and Chhim 2008)

• (Page 14 & 15- (Heuveline and Clauge 2024); (LA Brown et al. 2018); (DE Hinton et al. 2012)

• Page 18; (Bryan and Corso 2011)

Methodology

Participant

Clarity and Consistency of Numbers

• The author wrote, “A sample of 343 female, Cambodian participants...” but later you described Dataset 1 (160) + Dataset 2 (183) = 343 (correct).

• However, you also said “Dataset 2 consisted of 228 individuals…183 female participants were included.”

• Perhaps the author can clarify early that only female participants were included from Dataset 2, so readers immediately understand how you arrived at the final number.

Inclusion / Exclusion Criteria

• Dataset 1 has clear inclusion criteria.

• Please also state inclusion/exclusion criteria for Dataset 2 (e.g., age range, mental health condition, language ability). Otherwise, readers may wonder if both datasets are comparable.

Sampling Method & Size

• Briefly describe the sampling approach for both datasets (convenience sampling, purposive sampling, random selection?). This helps readers judge generalizability.

• How did you determine the sample size?

Demographic Description

• Consider reporting key demographic information (e.g., mean age, range, occupation, marital status). This will help readers understand the characteristics of your sample.

Instrument

Perhaps the author can briefly list the trauma exposure list such as parental mental health, physical abuse, emotional abuse in the Adverse Childhood Event checklist so it helps the reader to understand the origin of these predictive variables.

---

## [Reviewer Report]

This manuscript addresses a critical gap in global health by examining the relationship between exposure and experiences with PTSD severity, symptom clusters, trauma types, and suicide among Cambodian women. This topic is both relevant and timely today, focusing on social recovery and adjustment in the post-Khmer Rouge era. While the current paper has strengths, it could be improved through deeper cultural contextualization and theoretical integration with Cambodian scholarship. Below are specific comments and suggestions for improvement.

1.

Theoretical Weaknesses

a.

Limited Application of Indigenous Frameworks and Narratives: The paper would be strengthened by favoring indigenous frameworks over Western-centric psychiatric models (e.g., DSM-5 criteria for PTSD, PHQ-9 for depression) that may not fully capture Cambodian cultural expressions of distress or Cambodian-specific constructs (e.g., Nou’s “thinking too much” syndrome and Chhim’s baksbat).

b.

Underdeveloped Cultural Contextualization: Despite acknowledging the Khmer Rouge history, the paper does not theoretically address cultural contextualization and Cambodian trauma scholarship (e.g., intergenerational and collective memory, and healing factors), both of which are essential to Cambodian identity and recovery. Topics that could be beneficially explored include the role of Buddhist beliefs, karma, and spiritual healing practices in shaping and influencing mental health experiences, perceptions, and understandings.

c.

Neglect of Structural and Social Determinants: The overemphasis on individual psychological factors in Cambodian health research often overlooks the critical influence of structural determinants such as poverty, limited access to healthcare, and political and gender-based violence and collective identity. This lack can obscure the complex interplay between individual experiences and broader systemic oppression, potentially skewing research findings and undermining the effectiveness of healthcare interventions.

2.

Methodological Weaknesses

a.

IRB Approval, Consent, Confidentiality, and Protection of Human Subjects: While this study is essential, a foundational concern is the lack of formal institutional review board (IRB) approval from the author(s)’ home institution(s) in the United States. There is no discussion of the IRB process or the protection of human subjects, including the principles of anonymity and confidentiality. Additionally, clarification must be made of the rationale for seeking ethical approval from the Vietnam National University (VNU) and the National Ethics Committee for Health Research in Cambodia, rather than the standard IRB approval from the author(s)’ respective home institutions. Given that government control over local educational institutions can impact ethical guidelines and the protection of human subjects, to ensure informed consent and adequate safeguarding of participants, the author(s) must state whether the study received IRB approval in the United States.

b.

Measurement Validity and Cultural Fit: Although Khmer translations were applied, not all scales were validated for Cambodian populations, which raises validity concerns. Although the PHQ-9 and PCL-5 demonstrated strong internal consistency and reliability for both measures, further explanation is needed for their use compared to the more commonly employed Cambodian-tested measures, such as the Khmer version of the Hopkins Symptom Checklist-25 or the Harvard Trauma Questionnaire (among others). Additionally, it would help to understand why these scales better capture psychological distress and suicidal ideation. A Western or Eurocentric-focused symptom checklist may overlook or fail to fully capture the somatic expressions of distress, including social stressors and coping mechanisms. This concern is confirmed as stated on page 17, “Due to limited research in international and all-female populations, not all measures that were used in this study have been validated in Cambodia yet…”

c.

Stigma Attached to Reporting of Suicidal Ideations: It might be helpful to augment the quantitative findings with qualitative findings that capture Cambodian narratives of distress and symptomatology and suicidal ideations; using the single-item suicidal ideation assessment, PH-9, may not fully capture culturally nuanced expressions of suicidal thoughts. (See for example Nou, L. (2024). “Violence and traumatic stress among Cambodian survivors and perpetrators of the Khmer Rouge genocide.” Social Science Medicine - Mental Health, 6, 100341. https://doi.org/10.1016/j.ssmmh.2024.100341.)

d.

Sampling Bias: The small sample focused on entertainment workers and treatment-seeking female respondents and thus may not represent the broader experience of Cambodian women. This highlights the need for a wider discussionon how this population differs from mainstream Cambodian women. (For example, many women in the entertainment industry are of Vietnamese descent, which needs to be clarified for the reader to avoid generalizing between Vietnamese and Cambodian women.) Analyzing the demographics to identify independent and dependent variables may provide insights into both the commonalities and differences in the women’s experiences.

e.

Representative Sampling: In addition to considering who these women represent within the Cambodian population, it would be helpful to know the distribution of PHQ-9 scores for values 1, 2, and 3, specifically for the last question (9). It’s essential to combine values 2 and 3, as a score of 3 indicates thoughts of suicide every day. Therefore, before collapsing these scores into a primary dependent variable for non-linear multiple regression analyses (categorical), the authors would ideally provide additional details, especially if there are low scores for 3 or if the distribution is split between values 2 and 3. It would also be helpful to include the percentage of each score in the same line before merging them into a dichotomous dependent variable, as mentioned in lines 218 and 219 of the article.

f.

Retrospective Self-Report Bias: The Adverse Childhood Experiences (ACE) scale relies on memory recall, which can be distorted by a reluctance to “shame the family.” This desire to protect family reputation (saving face in Khmer culture) may lead to the inaccurate reporting of personal experiences.

g.

Pretesting or Pilot Testing the Scales: To fully reinforce the findings and ensure cultural integrity, it would be beneficial to consider pretesting and pilot testing the

scales. Additionally, holding focus group discussions on the concepts intended for the study could help address many of the concerns noted above.

3.

Culturally Grounded Critique, Limitations, and Implications

a.

Stigma and Disclosure: Cambodian cultural norms and mores discourage open dialogue about mental illness and suicide, which may affect both data accuracy and help-seeking behaviors. Survey methodologies and interventions must be adapted to Cambodian cultural norms and mores, including indirect approaches to suicidality and the integration of community-based healing practices (e.g., spirit mediums, family rituals).

b.

Mental Health Literacy: Ensuring conceptual and linguistic equivalencies is critical to ensure cultural sensitivity and understanding of concepts and theories, including the purpose. Pretesting and piloting, as noted, could help address these concerns.

c.

Mixed Methods: As suggested earlier, many of the suggestions could be addressed by including both quantitative and qualitative methodologies, reinforced by the literature and findings.

d.

Examine Gender/Feminist-Based Literature: Cambodia’s patriarchal structures, embedded in various social institutions, often shape the gendered experiences of Cambodian women, including how they express and narrate trauma. Thus, examining gender- or feminist-based literature becomes especially critical.

e.

Examine the Literature on Intergenerational Trauma and Community-Based Healing and Social Recovery: Linking the findings to the literature on

transmission of intergenerational trauma and community-based healing and social recovery will help provide contextualization and directions for future research.

f.

Policy Development: The findings should guide national health policies, foster collaboration with local mental health providers and institutions, and incorporate traditional and family-based healing approaches that are rooted in Cambodian beliefs and practices.

4.

Limitations and Recommendations for Future Research: I recommend addressing the concerns detailed above and incorporating a discussion of limitations and recommendations where appropriate and relevant.

---

## [Editor Report]

Dear Dr. Ean, 

I have read your manuscript and taken into consideration the reviewer comments. We would like to offer you the opportunity to respond to reviewer comments in a major revision of your manuscript. Please take special note of the comment below from Reviewer 2. Ethical approval for this research is a requirement for publication in Global Mental Health. Thank you for your interest in publishing your manuscript in this special issue of Global Mental Health. 

Comment from Review 2 regarding ethics approval:

IRB Approval, Consent, Confidentiality, and Protection of Human Subjects: While this study is essential, a foundational concern is the lack of formal institutional review board (IRB) approval from the author(s)’ home institution(s) in the United States. There is no discussion of the IRB process or the protection of human subjects, including the principles of anonymity and confidentiality. Additionally, clarification must be made of the rationale for seeking ethical approval from the Vietnam National University (VNU) and the National Ethics Committee for Health Research in Cambodia, rather than the standard IRB approval from the author(s)’ respective home institutions. Given that government control over local educational institutions can impact ethical guidelines and the protection of human subjects, to ensure informed consent and adequate safeguarding of participants, the author(s) must state whether the study received IRB approval in the United States.

Sincerely, 

Kristin Kosyluk, Guest Editor

---

## [Editor Report]

Dear authors,

Thank you for your resubmission. I have reviewed your responses to reviewer comments and appreciate how thoroughly you addressed most of the reviewer concerns. Before we can consider accepting this manuscript for the special issue, we need you to clarify the ethics approvals that were obtained for this research. 

One reviewer noted: 

IRB Approval, Consent, Confidentiality, and Protection of Human Subjects: While this study is essential, a foundational concern is the lack of formal institutional review board (IRB) approval from the author(s)’ home institution(s) in the United States. There is no discussion of the IRB process or the protection of human subjects, including the principles of anonymity and confidentiality. Additionally, clarification must be made of the rationale for seeking ethical approval from the Vietnam National University (VNU) and the National Ethics Committee for Health Research in Cambodia, rather than the standard IRB approval from the author(s)’ respective home institutions. Given that government control over local educational institutions can impact ethical guidelines and the protection of human subjects, to ensure informed consent and adequate safeguarding of participants, the author(s) must state whether the study received IRB approval in the United States. 

You responded to this as follows: 

The research reported in this manuscript is a secondary analysis of previously collected data. This data was collected by local researchers in Cambodia as part of doctoral work during the senior (last) author’s time at Vietnam National University. Therefore, the IRB approval that was received for this study reflects appropriate ethical review given the home institution of the individual’s collecting the data. This data was later shared with the other authors as part of ongoing research collaborations. No institutes in the US were involved in the collection of the data. 

Typical procedures to assure privacy and confidentiality were taken, including assigning random IDs to each participant, storing all collected data securely at Royal University of Phnom Penh, and training all individuals involved in the collection of data in ethical principles of research. 

We appreciate that this is a secondary analysis of existing data; however, we would still need you to include a statement about ethics approval for the original data collection. 

We suggest one of the following if these are accurate statements:

The analysis was performed on existing, publicly available secondary data that is fully anonymous (confirm that last part) 

The data used for this secondary analysis were collected as part of a previous research study: [NAME OF STUDY/IRB PROTOCOL, REVIEWING BODY, APPROVAL NUMBER].

If you can add one of these statements to your manuscript, we will be happy to consider it further. 

Best, 

Kristin Kosyluk

Guest Editor

---

## [Editor Report]

Dear Dr. Fleddermann and colleagues, 

Thank you for including the information on ethical approval for the research in your manuscript. We are now pleased to accept your manuscript for publication in the special issue. Congratulations, and thank you for your important work to understand suicidal ideation among Cambodian women. 

Warm regards, 

Kristin Kosyluk

Guest Editor